# Combination Antiretroviral Therapy and Immunophenotype of Feline Immunodeficiency Virus

**DOI:** 10.3390/v15040822

**Published:** 2023-03-24

**Authors:** Jeffrey Kim, Elisa S. Behzadi, Mary Nehring, Scott Carver, Shannon R. Cowan, Megan K. Conry, Jennifer E. Rawlinson, Sue VandeWoude, Craig A. Miller

**Affiliations:** 1Comparative Medicine Research Unit, School of Medicine, University of Louisville, Louisville, KY 40292, USA; 2Department of Microbiology, Immunology, and Pathology, College of Veterinary Medicine and Biomedical Sciences, Colorado State University, Fort Collins, CO 80523, USA; 3School of Natural Sciences, University of Tasmania, Hobart, TAS 7001, Australia; 4Department of Veterinary Pathobiology, Oklahoma State University, Stillwater, OK 74078, USA; 5Department of Clinical Sciences, College of Veterinary Medicine and Biomedical Sciences, Colorado State University, Fort Collins, CO 80523, USA

**Keywords:** antiretroviral therapy, lentiviral therapy, feline immunodeficiency virus, dolutegravir, tenofovir disoproxil fumarate, emtricitabine, immunophenotype, viral load

## Abstract

Feline Immunodeficiency Virus (FIV) causes progressive immune dysfunction in cats similar to human immunodeficiency virus (HIV) in humans. Although combination antiretroviral therapy (cART) is effective against HIV, there is no definitive therapy to improve clinical outcomes in cats with FIV. This study therefore evaluated pharmacokinetics and clinical outcomes of cART (2.5 mg/kg Dolutegravir; 20 mg/kg Tenofovir; 40 mg/kg Emtricitabine) in FIV-infected domestic cats. Specific pathogen free cats were experimentally infected with FIV and administered either cART or placebo treatments (*n* = 6 each) for 18 weeks, while *n* = 6 naïve uninfected cats served as controls. Blood, saliva, and fine needle aspirates from mandibular lymph nodes were collected to quantify viral and proviral loads via digital droplet PCR and to assess lymphocyte immunophenotypes by flow cytometry. cART improved blood dyscrasias in FIV-infected cats, which normalized by week 16, while placebo cats remained neutropenic, although no significant difference in viremia was observed in the blood or saliva. cART-treated cats exhibited a Th2 immunophenotype with increasing proportions of CD4^+^CCR4^+^ cells compared to placebo cats, and cART restored Th17 cells compared to placebo-treated cats. Of the cART drugs, dolutegravir was the most stable and long-lasting. These findings provide a critical insight into novel cART formulations in FIV-infected cats and highlight their role as a potential animal model to evaluate the impact of cART on lentiviral infection and immune dysregulation.

## 1. Introduction

Feline immunodeficiency virus (FIV) is a highly prevalent and currently untreatable lentivirus infecting domestic and feral cats worldwide. FIV is shed in saliva and is primarily transmitted via bite wounds. Infection leads to progressive immune dysfunction with clinical signs that include generalized lymphadenopathy, stomatitis or periodontitis [1], prolonged clotting times [2,3], and myelosuppression causing neutropenia, lymphopenia, and anemia [4,5,6]. FIV infection outcomes vary and can result in high morbidity and mortality in some cases [7], particularly in shelters and multi-cat housing situations [8]. Experimentally, certain strains of FIV have been found to result in high pathogenicity [9,10,11] and studies of FIV immunopathology and mechanisms of clinical symptoms have been utilized extensively to advance studies of human immunodeficiency virus (HIV), the cause of acquired immunodeficiency syndrome (AIDS) [12].

Combination antiretroviral therapy has been comprehensively studied as a treatment for primate immunodeficiency viruses (HIV, SIV), but there is a paucity of works in the literature investigating it as a treatment for FIV. Antiretroviral drugs used against HIV have also been shown to exhibit anti-FIV effects in vitro, including reverse transcription inhibitors such as tenofovir disoproxil fumarate [13,14] and emtricitabine [15,16,17,18]. The recent introduction of dolutegravir, a next-generation integrase inhibitor, is a preferred adjunctive treatment in HIV patients, causing a strong viral suppression of HIV with improved patient tolerance [19,20,21,22]. The synergy of multiple antiretroviral classes—often referred to as combination antiretroviral therapy (cART) or highly active antiretroviral therapy (HAART)—has demonstrated improved patient outcomes and is the gold-standard for antiretroviral therapy. As cART with tenofovir disoproxil fumarate, emtricitabine, and dolutegravir has demonstrated lentiviral suppression in humans and nonhuman primates [23,24,25,26,27,28,29,30], we thus developed a novel cART protocol [24,29,31] and evaluated its efficacy as a treatment for cats infected with FIV.

Previous studies investigating the treatment of FIV have demonstrated that antiretroviral drugs can effectively suppress FIV infection in culture [16,32,33]. Other antiretroviral drugs are also neuroprotective [34] and decrease the viral load in chronically infected cats [35]. Thus, a comprehensive analysis of cART’s impact on viral and proviral loads, salivary shedding, and clinical morbidity from acute to chronic infection may further aid in the design of cART. A complete assessment using highly sensitive quantitative techniques while investigating the immune response to infection and therapy will also improve our understanding and treatment of FIV. In this study, we inoculated cats with an immunopathogenic strain of FIV (FIV_c36_) [9,36] and treated FIV-infected and uninfected control cats daily with cART or a placebo. Clinical and hematologic parameters were analyzed, viral RNA and proviral DNA in blood and saliva were quantified by ddPCR, and lymphocyte immunophenotypes during infection and treatment were characterized by flow cytometry in blood and lymph nodes. In addition to cART pharmacokinetics, we also evaluated the impacts of cART on viremia and shedding as a treatment for individual FIV-patients.

This is the first study to assess the treatment efficacy of cART in cats experimentally infected with FIV using a novel daily dosing application. We present a comprehensive evaluation of cART as a treatment for FIV by assessing the clinical impacts, the reduction of viral RNA and proviral DNA in blood, the inhibition of salivary shedding, and immunophenotype characterization in response to cART. Our findings suggest that cART inhibits some FIV-induced myelosuppression, restores Th17 cells in the oral compartment, and triggers a Th2 immune response.

## 2. Materials and Methods

### 2.1. In Vivo Protocol

Eighteen cats (*n* = 18) from 6 to 10 months old, bred from a specific pathogen free colony, were utilized for the present study. The cats were housed in barrier rooms and fed ad libitum in accordance with Colorado State University IACUC-approved protocols at an AAALAC-international accredited animal facility. The cats were housed in groups of 3 to 4 animals, and control cats were housed separately from FIV-infected cats. Two weeks prior to the initiation of the study, the cats were habituated to handling for blood and saliva collection and physical examination. At day 0, *n* = 12 cats were intravenously and orally inoculated with 1 mL of FIV_c36_ viral stock diluted 1:80 with 0.9% saline, previously validated to demonstrate acute immunopathogenic and reproducible high viral titers [9,36]. The remaining *n* = 6 cats were similarly sham-inoculated with 1 mL of 0.9% saline as negative controls. Beginning at week 5 post-inoculation through week 24, *n* = 6 FIV-inoculated cats received daily combination antiretroviral therapy (cART) subcutaneously consisting of two reverse transcription inhibitors, tenofovir disoproxil fumarate (TDF 20 mg/kg/day) and emtricitabine (FTC 40 mg/kg/day), and an integrase inhibitor dolutegravir (DTG 2.5 mg/kg/day). The other *n* = 6 FIV-inoculated cats received 1 mL subcutaneous injections of 15% kleptose, the cART vehicle, daily. These three cat groups were referred to as control, cART-treated FIV+, and placebo-treated FIV+ cats, respectively. Figure 1 illustrates the distribution of cat groups. All cat groups were equally distributed with female-intact and male-neutered cats, and FIV infection and treatment assignments were randomized. Physical examination recordings were conducted for each cat at all blood and saliva collection timepoints, and included weight, temperature, heart rate, respiratory rate, mentation, oral cavity (i.e., mucous membrane color, capillary refill time, and gingival exam), lymph node palpation, and abdominal palpation (Appendix B).

### 2.2. Blood and Saliva Collection

Blood was collected from all cats prior to FIV inoculation on day 0, weekly post-inoculation from weeks 1 to 6, and then every other week from weeks 8 to 24. Up to 6 mL of blood was collected from a cephalic vein into ethylenediaminetetraacetic acid (EDTA) tubes and immediately placed onto a rocker. Complete blood count (CBC) analyses were assessed at baseline (day 0) and weeks 4, 6, 10, 16, and 24. Remaining EDTA-treated blood was aliquoted into 15 mL conical tubes and centrifuged at 900× *g* rpm for 10 min at 23 °C. Plasma and buffy coat were aliquoted into 1.5 mL microcentrifuge tubes.

Saliva was collected on weeks 2, 6, 12, 18, and 33. To collect saliva, each cat’s mouth and cheek pouches were swabbed using 4 sterile polyester swabs. Each swab was held in a cat’s mouth for 20–30 s. Swab tips were broken off into sterile 3 mL syringe barrels with the plungers removed and placed in sterile 15 mL conical tubes on ice. After collection, the conical tubes were centrifuged at 1500× *g* rpm for 10 min to transfer the saliva from the swabs into the conical tubes. Plasma, buffy coat, and saliva were all stored at −80 °C.

### 2.3. Viral RNA and Proviral DNA Extractions

Viral RNA was extracted from plasma for all timepoints using a QIAmpViral RNA Mini Kit (Qiagen, Valencia, CA, USA) according to the manufacturer’s instructions with minor modifications. The user-adapted protocol steps included: (1) Pipette 554.4 µL prepared Buffer AVL containing carrier RNA, along with 5.6 µL of carrier AVE, into a 1.5 mL microcentrifuge tube. (2) Continue following the manufacturer’s instructions, and once Buffer AW2 is added close the cap and centrifuge at 16,000× *g* for 4 min. (3) Place the QIAamp Mini column into a clean 2 mL collection tube and discard the tube containing the filtrate. Centrifuge at 16,000× *g* for 1 min. Place the QIAamp Mini column into a clean 1.5 mL microcentrifuge tube and discard the tube containing the filtrate. (4) Add 40 µL RNase-free water equilibrated to room temperature. Close the cap and incubate at room temperature for 1 min. (5) Centrifuge at 6000× *g* for 1 min. (6) Repeat steps 4 and 5. Viral RNA quality was assessed via spectrophotometry and converted to cDNA as previously described [37].

Proviral DNA was extracted from buffy coat for all timepoints using a DNeasy Blood and Tissue Kit (Qiagen, Valencia, CA, USA) according to the manufacturer’s instructions with minor modifications. The user-adapted protocol steps included: (1) Pipette 50 µL of buffy coat, and add 150 µL of Phosphate Buffered Saline (PBS, Fisher scientific, Waltham, MA, USA). (2) Continue following the manufacturer’s instructions, and once Buffer AW2 has been added centrifuge for 4 min at 16,000× *g*. Discard the flow-through and collection tube. (3) Transfer the spin column to a new 1.5 mL microcentrifuge tube. (4) Incubate Buffer EB at 56 °C for 4 min. Elute the DNA by adding 50 µL of the incubated Buffer EB to the center of the spin column membrane. Incubate for 1 min at room temperature. Centrifuge for 1 min at 6000× *g*. (5) Repeat steps 3 and 4. Extracted proviral DNA concentrations were immediately measured using a Qubit dsDNA HS Assay Kit (Invitrogen, Eugene, OR, USA) according to manufacturer recommendations.

Saliva was diluted with PBS to 330 µL total volume. Saliva proviral DNA and viral RNA were extracted as previously validated and described [38]. Viral RNA quality was assessed and converted to cDNA as described above. Saliva proviral DNA concentrations were measured as described above. All blood and saliva RNA, cDNA, and DNA samples were stored at −20 °C. One placebo-treated FIV+ cat was excluded from saliva collection due to its behavior to minimize the cat’s stress and ensure personnel safety.

### 2.4. Viral and Proviral Quantification

All samples were analyzed in duplicate. FIV-C proviral DNA was quantified via ddPCR using previously described protocols [39] and the following primers and probes:

Forward primer 5′-TGAGTCAGCCCTATCCCCATTA-3′;

Reverse primer 5′-ACTCACCCTCCTGATGGTCCTA-3′;

Probe 5′-/56-FAM/ACCATTGCC/ZEN/ATACTTCACTGCAGCCG/31ABkFQ/-3′.

PCR was performed in a CFX Connect^TM^ thermal cycler (Bio-Rad, Hercules, CA, USA) and droplets were cycled for 10 min at 95 °C, followed by 49 amplification cycles (30 s at 95 °C and 1 min at 58.8 °C), ending with 10 min at 98 °C. FIV-C viral RNA was similarly quantified via ddPCR with the following user-adapted protocol: (1) Create a master mix as described above but at a 10:1:4 ratio, respectively. (2) Add 15.9 µL of the master mix and 5.3 µL of the undiluted extracted viral cDNA into a 0.2 mL PCR tube, and continue as previously described [39].

The proviral copy number was normalized to copies per 10^6^ cells. Cell quantification per well was measured targeting CCR5 via ddPCR as described above, using the following primers and probes:

Forward primer 5′-ACGTCTACCTGCTCAACCTGG-3′;

Reverse primer 5′-ACCGTCTTACACATCCCATCCC-3′;

Probe 5′-/56-FAM/TCCGACCTG/ZEN/CTCTTCCTCTTCACCCTCC/31ABkFQ/-3′.

All primers and the probe were diluted to 9 µM and combined at a 0.44:1.11:0.44 ratio, respectively. The CCR5 ddPCR protocol utilized Supermix for Probes no dUTP (Bio-Rad #1863024) with the CCR5 primers/probe mixture at a 10:2 ratio, respectively. ddPCR steps were followed as described above for the proviral DNA analysis.

### 2.5. Flow Cytometry

Flow cytometry analyses were performed on blood collected at baseline and weeks 2, 4, 6, 10, 15, and 24. Mandibular lymph node fine needle aspirates using a 22 g needle were collected for flow cytometry analyses at baseline and weeks 4, 11, 16, and 24. Lymph node aspirates were placed into PBS with 10% fetal bovine serum (FBS) to obtain single cell suspensions. Two panels (“Lineage/FOXP3” and “Activation”) were utilized to stain 50 µL of EDTA-treated blood or approximately 5 × 10^5^ lymph node cells. Activation samples were surface stained with CD4 FITC (Fisher, clone 3-4F4), CXCR3 AF405 (Novus Biologicals, clone 49801), CCR4 R-PE (FabGennix, Frisco, TX, USA, K5 5 polyclonal conjugated in house with Biotium Mix *n* Stain kit), and CCR6 PerCP or PerCP Cy5.5 (R&D Systems clone 53103, BD Biosciences clone 11A9, respectively) at the manufacturer’s recommended volume per test for 20 min in the dark at 4 °C. Red blood cells were lysed and samples fixed using the TQ-Prep Workstation and IMMUNOPREP Reagent System (Beckman Coulter Inc., Brea, CA, USA). Lineage/FOXP3 samples were surface stained as above, but with CD4 FITC, CD8 PE (Southern Biotech, clone fCD8, Birmingham, AL, USA), CD21 AF647 (Bio-Rad, Hercules, CA, USA, CA2.1D6), followed by a wash with PBS + 1% FBS. Pellets were resuspended in eBioscience™ Foxp3/Transcription Factor Fixation/Permeabilization buffer (Invitrogen, Waltham, MA, USA) diluted according to the manufacturer’s recommendation and incubated in the dark at 4 °C for 1 h. Samples were washed in 1× Permeabilization buffer and resuspended in anti FOXP3 PECY7 (Fisher, clone FJK-16s) diluted in 1× Permeabilization buffer. An incubation at 4 °C for 30 min in the dark was followed by two washes in 1× Permeabilization buffer and lysing/fixation as described above. Unstained, single-stained and pertinent FMO controls were prepared for each experiment. Data were acquired using BD FACSDiva™ Software interfaced with a BD FACSAria™ SORP instrument (Becton Dickinson, San Jose, CA, USA). Compensation values were determined using single stained controls. Gating proceeded from singlets to lymphocytes to CD4, CD8, or CD21, and CD4 cells were examined for each activation marker. The percentage of lymphocytes positive for each marker was evaluated over time and compared to baseline values and naïve control data to compare alterations in the lymphocyte immunophenotype in response to cART treatment and in the presence of FIV infection. Immunophenotype cell counts were calculated as previously described [36,40,41] and compared with CBC and ddPCR data to evaluate changes in the circulating immunophenotypes compared to FIV viral and proviral loads over the course of the study and at individual time points.

### 2.6. Pharmacokinetics

A pharmacokinetic study was conducted to confirm the appropriate absorption of the cART drugs. Six FIV-infected cats were administered a single cART dose, and 1 mL of blood was collected at 0.5, 1, 2, 4, 8, and 24 h after cART administration. Liquid chromatography–tandem mass spectrometry and Sciex Analyst^®^ 1.7.1 software (Framingham, MA, USA) with HotFix using Analyst Classic were used to determine the plasma cART concentrations.

### 2.7. Ethics Statement

This study was approved by the Colorado State University Institutional Animal Care and Use Committee; 1142—Impacts of antiretroviral therapy on oral cavity homeostasis in an FIV animal model. The Colorado State University animal care program is licensed by the United States Department of Agriculture, maintains an Office of Laboratory Animal Welfare Public Health Service assurance (A3572-01), and is accredited by the Association for Assessment and Accreditation of Laboratory Animal Care International. Any animal that exhibited clinical signs of FIV infection or other morbidities was evaluated and treated by a clinical veterinarian. No animal was euthanized due to illness.

### 2.8. Statistical Analysis

Mixed effects Generalized Additive Models (GAM) were used to evaluate how physical, viral and immune response variables differed among the treatment groups (control, cART, placebo) on average and over time, with cat ID as a random effect to accommodate the repeated measures. GAMs were specifically selected in this study to account for the frequent non-linear temporal dynamics in response variables, and to provide superior capacity to evaluate treatment differences over time compared to common linear methods (e.g., repeated measures ANOVA). Within each GAM, treatment was included as a factorial fixed effect to examine average differences, and an interaction term between treatment and time (week) as a spline factor was also included to evaluate whether each treatment group changed over time. A post hoc test was used to determine whether the temporal splines differed among the treatment groups. Prior to all analyses, the distribution of the response variable was assessed and, where relevant, log10 transformed to normalize the data. All analyses were undertaken in Rv4.0.3 using the packages ‘mgcv’ and ‘car’. Results of statistical analyses are presented in Appendix A.

## 3. Results

### 3.1. cART Improves the Hematologic Impact of FIV-C Infection

Hematologic changes were most notable among neutrophils with significant differences between cART-treated and placebo-treated FIV+ cats (*p* < 0.001; Figure 2A,B and Appendix A). Mean neutrophil values in cART-treated FIV+ cats normalized by week 16, but placebo-treated FIV+ cats remained neutropenic at all timepoints post-infection. cART-treated FIV+ cats had lower lymphocyte values compared to placebo-treated FIV+ cats, but lymphopenia was not observed.

### 3.2. Impact of cART on FIV Viral and Proviral Loads

Viral RNA and proviral DNA were detected in the blood and saliva of all FIV-infected cats over the course of this study, whereas none were detected in uninfected control cats. Infection peaked at week 2, with mean blood viral loads of 3.38 × 10^7^ copies/mL among cART-treated FIV+ and 1.08 × 10^7^ copies/mL among placebo-treated FIV+ cats (Figure 3C,D). Mean blood proviral loads were 3.26 × 10^5^ copies/10^6^ cells among cART-treated FIV+ cats and 4.94 × 10^4^ copies/10^6^ cells among placebo-treated FIV+ cats (Figure 3A,B). Blood viral (Figure 3C,D) and proviral loads (Figure 3A,B) steadily declined afterwards and were not significantly different throughout the study (Appendix A).

Mean viral RNA loads in saliva were lower in cART-treated FIV+ cats compared to placebo-treated FIV+ cats, but they were similar from week 6 to week 33 (Figure 3G,H, Appendix A). Viral RNA peaked in cART-treated FIV+ cats at week 6 with 5.29 × 10^3^ copies/mL, and in placebo-treated FIV+ cats at week 2 with 1.03^5^ copies/mL (Figure 3G,H). Mean salivary proviral DNA peaked in cART-treated FIV+ cats and in placebo-treated FIV+ cats at week 6, with 1.77 × 10^5^ copies/mL and 6.96 × 10^4^ copies/mL, respectively (Figure 3E,F). Proviral DNA remained relatively stable for the remainder of the timepoints and was not significantly different (Figure 3E,F, Appendix A).

### 3.3. cART Impacts on Lineage and Activation Markers

CD4^+^ T cells declined in both the placebo-treated and cART-treated FIV+ cats relative to the uninfected control cats (Figure 4A,B). Notably, CD4^+^ T cells declined markedly for 10 weeks following the initiation of cART treatment, followed by a rebound to levels comparable to placebo-treated FIV+ cats (Figure 4A,B). There were no differences in the proportion of CD8^+^ cells, CD21^+^ cells, B:T cell ratio, or the CD4^+^:CD8^+^ T cell ratio in blood between the two groups (Appendix A). There was also a significant difference over time in the proportion of CD4^+^CCR4^+^ cells in blood between all treatment groups (*p* < 0.001; Figure 4C,D, Appendix A). This cell type was highest in cART-treated FIV+ cats, while placebo-treated cats had higher proportions compared to control cats. No differences in the proportion of CD4^+^CXCR3^+^ or CD4^+^CCR6^+^ cells were detected over time (Appendix A).

In contrast to blood, the proportion of CD21^+^ cells obtained from the mandibular lymph node was lower in cART-treated FIV+ cats compared to placebo-treated FIV+ cats (*p* < 0.001), but was similar after cART treatment began (Appendix A). However, there were no differences in CD4^+^, CD8^+^, B:T cell ratio, and the CD4^+^:CD8^+^ T cell ratio (Appendix A) between groups. The number of CD4^+^CCR6^+^ cells increased in cART-treated FIV+ cats since starting treatment compared to placebo-treated FIV+ cats (*p* < 0.001, Appendix A), but they were equalized by week 24 (Figure 5A,B). No differences were observed among the proportion of CD4^+^CXCR3^+^ and CD4^+^CCR4^+^ cells for any of the groups (Appendix A).

### 3.4. Pharmacokinetics

Plasma cART concentrations were determined across 24 h for female-intact and male-neutered cats. Tenofovir plasma concentrations (C_max_ 1230–2120 ng/mL) peaked at 0.5 h; emtricitabine plasma concentrations (C_max_ 22,500–32,400 ng/mL) peaked at 1 h; and dolutegravir plasma concentrations (C_max_ 1290–2260 ng/mL) peaked at 2 h (Figure 6).

## 4. Discussion

Despite excellent therapies being available for human and simian immunodeficiency viruses, a reliable antiviral treatment for FIV remains to be established. A cART protocol was previously shown to suppress viremia in nonhuman primates infected with simian immunodeficiency viruses [24,29,31], a lentivirus similar to FIV, and we investigated the cART protocol, doses, and efficacy in reducing viremia and treating FIV in experimentally infected cats. To elucidate the impact of cART we evaluated hematologic changes, quantified viral RNA and proviral DNA in blood and saliva, and characterized the immunophenotype of infection and cART. Our findings demonstrate that: (1) the presented cART protocol improves neutrophil counts, suggesting that cART has some inhibition of FIV-induced myelosuppression; (2) FIV causes a greater Th2 immunophenotype compared to Th1, as evidenced by the increased proportion of CD4^+^CCR4^+^ cells; and (3) dolutegravir was the most stable and long-lasting, and it may be the most promising of the three cART drugs to treat FIV infection.

Primary clinical signs and hematologic impacts of FIV infection include generalized lymphadenopathy, stomatitis or periodontitis [1], prolonged clotting times [2,3], and myelosuppression causing neutropenia, lymphopenia, and anemia [4,5,6]. Our cats demonstrated successful infection from FIV-C with infectivity similar to a previous experimental FIV model, with analogous patterns in neutrophil, FIV provirus, and CD4^+^ changes [36]. Considering experimental infection, the clinical impacts of FIV infection were minor, primarily including mild peripheral lymphadenopathy while lacking the pyrexia and lethargy that are commonly associated with acute FIV infection [6,7,42]. Neutropenia was the most notable change, and placebo-treated FIV+ cats remained neutropenic, as previously observed [36], while cART-treated FIV+ cats eventually normalized, suggesting that cART has some inhibitory effect on FIV infection’s myelosuppression. Although there were significant differences of CD4^+^ lymphocytes in the blood between cART and placebo-treated FIV+ cats, no lymphopenia was appreciated. Lymphopenia is commonly observed in FIV-infected cats due to an FIV-induced cytopathic impact from a direct replication in CD4^+^ lymphocytes [6,12], but we may not have observed lymphopenia since it is more frequently seen in younger animals, during chronic infection, and in clinically ill cats [6].

The presented study is the first to investigate the proposed cART protocol as a treatment for FIV-infected cats, and its impacts on viremia were evaluated using droplet digital PCR (ddPCR). Diagnostic assays to study lentiviruses have greatly improved with advanced quantitative techniques such as ddPCR [43,44,45,46,47] due to its high precision and reproducibility [48,49]. ddPCR has rarely been used to quantify FIV loads [50], presenting a valuable opportunity to apply its high sensitivity to evaluate therapeutic responses during infection. We observed that, prior to cART, infected cats had high levels of FIV viral RNA and proviral DNA, as previously reported [36,38]. The present study’s cART protocol was developed because other studies demonstrated viral suppression in rhesus macaques infected with simian immunodeficiency virus using a similar [23,24,25,26,27,28] or the same cART protocol [24,29,31]. Additionally, these reverse transcription inhibitors, namely tenofovir disoproxil fumarate [13,14] and emtricitabine [15,16,17,18], have also been shown to exhibit anti-FIV effects in vitro. However, cART did not suppress viremia significantly in vivo, despite cART-treated FIV+ cats exhibiting improved neutrophil quantities compared to placebo-treated FIV+ cats. One reason for inefficacy may be that the concentrations of circulating drugs were not adequate to exhibit direct antiviral activity. Dolutegravir plasma concentrations were the most stable of all cART drugs in our cats, suggesting that our dolutegravir dosing protocol may be an appropriate FIV therapy, although further investigation is warranted. However, while the pharmacokinetic profile of dolutegravir appears sufficient in FIV-infected cats, tenofovir and emtricitabine may require greater concentrations and durations for cART efficacy. Though all three drugs reached appropriate therapeutic concentrations for 24 h based upon primate and human studies, we cannot be certain that these compounds reached appropriate concentrations intracellularly in cats or whether they will ultimately interfere with replication.

Although cART-treated FIV+ cats had a significantly lower viral load in saliva compared to placebo-treated FIV+ cats, this was likely due to the differences observed prior to cART and the viral loads remained similar for most of the study duration.

To further characterize the immunophenotype of experimental FIV infection and cART, lineage and activation markers in blood and lymph nodes were assessed. Evaluating the infection and its response to cART in mandibular lymph nodes was especially important since FIV has significant tropism for oral lymphoid tissue and may be an initial site for circulating virus to infect resting T lymphocytes [38]. Initial infection with the FIV_C36_ strain typically causes a transient increase in CD4^+^ T lymphocytes [40], followed by a shift to CD8^+^ cells and a gradual decline in CD4^+^ cells causing a characteristic inversion of the CD4^+^:CD8^+^ ratio [6,51,52,53,54,55]. Changes in CD4^+^ followed a similar pattern as previously seen [36], but we did not observe the rapid initial increase in CD4^+^ T cells after infection that is typically accompanied by sharp increases in viral and proviral loads [7,56]. Additionally, contrary to our predictions, cART did not improve CD4^+^ T cells in the blood of treated FIV+ cats, and this immunophenotype was in fact significantly lower in cART-treated FIV+ cats than in placebo-treated FIV+ cats between weeks 5 and 16. Similarly, the proportion of CD21^+^ cells, a broad marker for B cells, was significantly lower in cART-treated FIV+ cats compared to placebo-treated FIV+ cats. However, this was likely due to proportional differences prior to the initiation of cART, as the proportion of CD21^+^ cells did not differ between cART-treated and placebo-treated FIV+ cats once treatment began. While the cause for these discrepancies in lymphocyte lineage is unknown, these findings highlight a potential impact of cART on overall immune function. Deficiencies in resting CD4^+^ T cells may cause a predisposition to opportunistic infection, especially in the oral cavity and gastrointestinal tract [38,57,58,59], while at the same time they may reduce the viral burden by decreasing circulating target cells required for active viral replication. Future studies investigating the direct consequences of cART on these critical lymphocyte immunophenotypes may, therefore, have broad implications in both human and animal lentiviral infections.

We further evaluated the immunologic response during FIV infection by characterizing Th1, Th2, and Th17 lymphocyte subtypes. CXCR3 is a marker of the Th1 subtype of lymphocytes [60,61,62,63,64], which are generally proinflammatory and can perpetuate an autoimmune response [65]. Neither FIV infection nor cART appeared to impact Th1-type cytokines, since there were no differences in the proportion of CD4^+^CXCR3^+^ lymphocytes in blood and lymph nodes between any treatment group across the study. There were, however, notable differences across all treatment groups in the proportion of circulating CD4^+^CCR4^+^ lymphocytes, a marker of Th2 subtypes [64,66] which are associated with atopy and/or an anti-inflammatory response [65,67]. Naturally, the proportion of CD4^+^CCR4^+^ lymphocytes in placebo-treated cats was slightly higher than in uninfected control cats, since FIV-infected cats will mount some Th2 anti-inflammatory response to infection [37,68]. However, in cART-treated FIV+ cats, the proportion of CD4^+^CCR4^+^ cells was significantly higher than in placebo-treated FIV+ cats. The significance of this cART-induced shift in the Th2 immunophenotype is unclear, but it suggests that cART may suppress inflammation and immune activation due to an upregulated Th2 immunophenotype [65], or may impact viral clearance due to its role in inhibiting antiviral Th1 responses [69]. Although no data were found in the timeframe of this study to support the latter, our results support Dean and Pedersen’s conclusions that FIV infection causes a greater Th2-type response compared to Th1 [70], a distinction from other reports that do not demonstrate a clear Th1 to Th2 shift [68,71,72,73,74].

There was also an increase in CD4^+^CCR6^+^ lymphocytes in the mandibular lymph nodes of cART-treated FIV+ cats. CCR6 are a marker of the Th17 lymphocyte subtype [61,75,76], which are also proinflammatory cells [77] that can support FIV replication [78,79]. Th17 cells and IL-17 receptor signaling are essential for mucosal host defense against pathogens, and a decrease in Th17 cells due to HIV and FIV infection has been attributed to the development of lentiviral-induced oral disease [12,38,80]. The proportion of CD4^+^CCR6^+^ cells in lymph node aspirates of cART-treated FIV+ cats significantly increased to similar proportions seen in uninfected control cats, indicating that cART may rescue this vital lymphocyte subtype and restore mucosal immunity. There was a transient decrease in circulating (blood) CD4^+^CCR6^+^ cells in placebo-treated FIV+ cats at weeks 11 and 16, suggesting a decreased inflammatory response as they transitioned from acute to chronic infection. Collectively, these results suggest that FIV primarily induces a Th2 immunophenotype compared to Th1 and Th17, and the presented cART protocol may promote an even greater Th2 response and partially restore Th17 cells in the oral compartment.

All three drugs in cART reached high plasma concentrations, but it is possible that the drugs did not reach sufficient intracellular concentrations and or the duration of activity to interfere with proviral load and viral replication. Tenofovir plasma concentrations reached a C_max_ considerably higher than simian immunodeficiency virus-infected macaques [81], followed by a rapid decline. In humans receiving chronic therapy, the median tenofovir trough plasma concentration was 50 ng/mL (35 to 77 ng/mL) 22 to 26 h after drug exposure [82]. Similarly, humans receiving tenofovir 300 mg every 48 h or 150 mg every 24 h had median minimum plasma concentrations of 40 ng/mL (8 to 100 ng/mL) or 70 ng/mL (30 to 140 ng/mL), respectively [83]. In this study, all cART-treated cats (6/6) reached plasma concentrations below 40 ng/mL by 8 h. Like tenofovir, emtricitabine plasma concentrations were considerably higher in cats compared to macaques [84], and exceeded human blood levels for at least 8 h (C_max_ 1820 ng/mL; C_min_ 47 ng/mL) [85]. Dolutegravir’s C_max_ measured in this study was similar to humans’ and macaques’ [86,87]. Its trough levels of 409 to 1260 ng/mL were higher than the plasma levels reported in macaques [88], and within the range of levels attained in humans [86,89,90]. Furthermore, dolutegravir concentrations were maintained above the purported target minimum concentration of 64 ng/mL [22,91] at all timepoints.

The results of this study demonstrate new evidence of cART reducing myelosuppression of FIV. The rebound of circulating neutrophil levels after cART administration illustrates that this protocol inhibits chronic neutropenia and associated immune dysfunction. The restoration of Th17 cells in the oral compartment suggests that cART represents a plausible therapy to reduce or eliminate FIV-induced oral disease and concomitant transmission. Among the three cART drugs investigated, dolutegravir may be the most promising treatment since it was the most stable and long-lasting. Collectively, our results demonstrate that refinement of the cART protocol may have positive benefits for cats with chronic FIV immune dyscrasias.

## Figures and Tables

**Figure 1 viruses-15-00822-f001:**
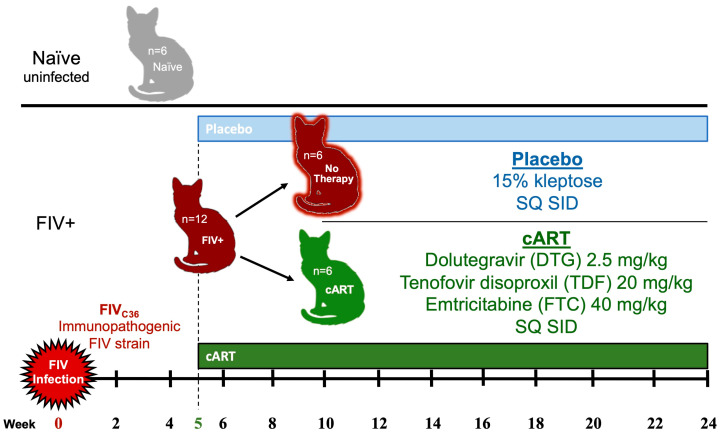
Distribution of cats and treatment assignments. Eighteen cats were randomly distributed across three groups, equally containing female-intact and male-neutered cats. At day 0, twelve cats were inoculated with FIV_C36_ and one group (*n* = 6) was treated daily with cART starting at week 5, while the other group (*n* = 6) received placebo. The remaining were naïve uninfected control cats (*n* = 6).

**Figure 2 viruses-15-00822-f002:**
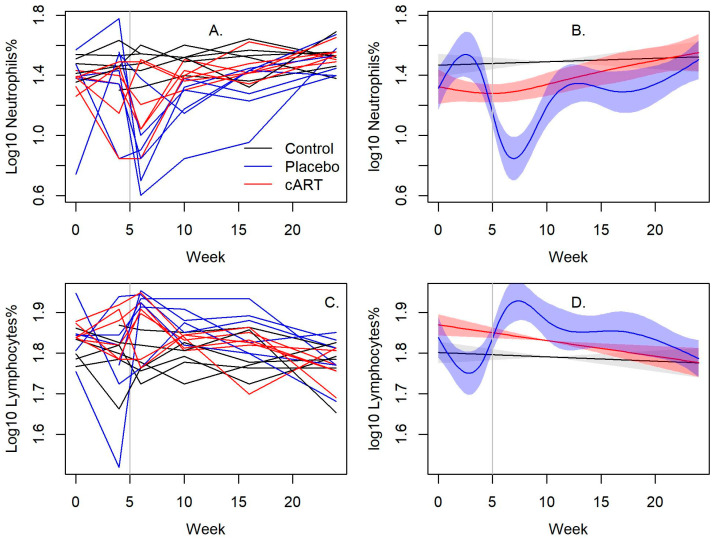
cART reduces myelosuppression of FIV infection. Neutrophil values in FIV+ cats treated with cART normalized by week 16, while placebo-treated FIV+ cats were persistently neutropenic (*p* < 0.001). Lymphocyte values were normal in all groups but were lower in cART-treated FIV+ cats compared to placebo-treated FIV+ cats. The vertical bar indicates when cART and placebo treatment began. (**A**,**C**) Line plots of individual cats over time and (**B**,**D**) their associated spline fits from GAM analyses are presented.

**Figure 3 viruses-15-00822-f003:**
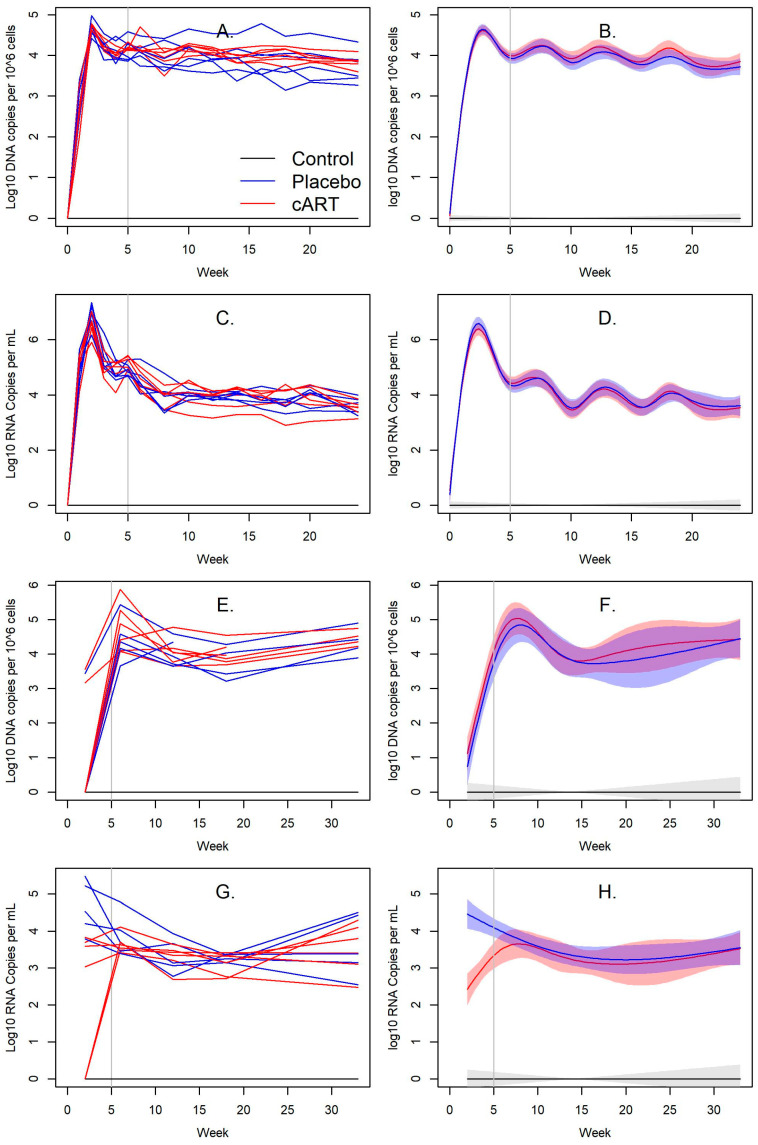
cART reduces salivary viral RNA loads in FIV+ cats. (**A**–**D**) Blood FIV viral RNA and proviral DNA loads did not differ between cART-treated FIV+ cats and placebo-treated FIV+ cats. (**E**–**H**) Viral RNA loads in the saliva of cART-treated FIV+ cats were lower than placebo-treated FIV+ cats, but they remained similar starting from week 6. The vertical bar indicates when cART and placebo treatment began. (**A**,**C**,**E**,**G**) Line plots of individual cats over time and (**B**,**D**,**F**,**H**) their associated spline fits from GAM analyses are presented.

**Figure 4 viruses-15-00822-f004:**
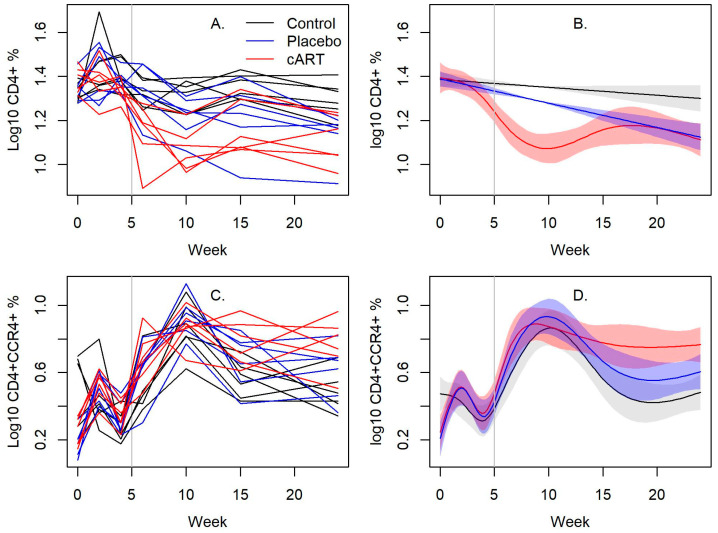
FIV infection induces a primary Th2 immunophenotype. cART significantly lowered CD4^+^ T cells in the blood of FIV+ cats compared to those treated with a placebo. CD4^+^CCR4^+^ cells in cART-treated FIV+ cats were significantly higher than in placebo-treated FIV+ cats, and both were slightly higher than in uninfected control cats. cART may inhibit inflammation and immune activation of FIV infection with an upregulated Th2 immunophenotype. The vertical gray bar indicates when cART and placebo treatment began. (**A**,**C**) Line plots of individual cats over time and (**B**,**D**) their associated spline fits from GAM analyses are presented.

**Figure 5 viruses-15-00822-f005:**
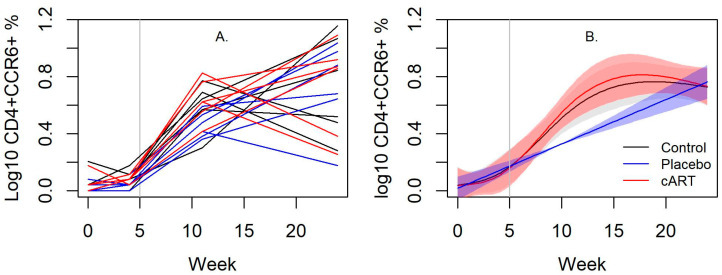
cART improved oral immunity in FIV infected cats. In the mandibular lymph nodes of cART-treated FIV+ cats, CD4^+^CCR6^+^ cells were significantly higher than in placebo-treated FIV+ cats. The increase of CCR6^+^ lymphocytes in cART-treated FIV+ cats may illustrate a partial restoration in Th17 cells. Collectively, the results may demonstrate that cART improves mucosal immunity. The vertical bar indicates when cART and placebo treatment began. (**A**) A line plot of individual cats over time and (**B**) its associated spline fit from GAM analyses are presented.

**Figure 6 viruses-15-00822-f006:**
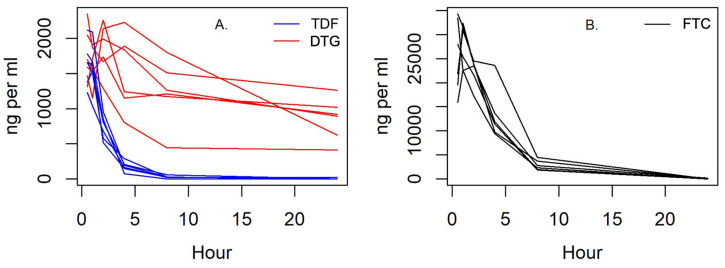
Dolutegravir (A) plasma concentration remained stable for 24 h. (**A**) Tenofovir (TDF) and (**B**) emtricitabine (FTC) plasma concentrations quickly declined in cats and remained low by 4 and 8 h after administration, respectively.

## Data Availability

Not applicable.

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
