# Peer review of "Combination Antiretroviral Therapy and Immunophenotype of Feline Immunodeficiency Virus"

_viruses, 2023, doi:10.3390/v15040822_

Round 1
Reviewer 1 Report
This is a very interesting paper addressing existing gap of knowledge. I have enjoyed reading it. Thank you for your contribution to feline retrovirology.
Author Response
Thank you for the kind words and your review of our manuscript
Reviewer 2 Report
Feline immunodeficiency is an important viral disease in domestic cats caused by chronic and persistent infection by a lentivirus (FIV). At present, there is no definitive therapy to treat infected cats, although several studies have been conducted using different antiretroviral compounds used against human immunodeficiency virus (HIV), none of which have been found to be very effective. Although there is evidence of the action of antiretroviral drugs against FIV both in cell culture and in animal infection, further studies are still lacking to analyse their effect on viral load, clinical and immunopathological parameters of infected animals. On the other hand, very little is known about the use of combination antiretroviral therapies (cART, HAART) in FIV-infected cats similar to what is done in HIV-infected patients.
In the manuscript, the authors evaluate the effect of cART (combined antiretroviral therapy), with three antiretroviral compounds previously used in HIV, in domestic cats experimentally infected with FIV. The main parameters studied are RNA viral and DNA proviral loads in blood and saliva, and lymphocyte immunophenotypes in blood and mandibular lymph nodes. The results are compared with those obtained in infected cats treated with placebo and uninfected cats. The authors conclude that their results seem to suggest that cART inhibits some FIV-induced myelosuppression, restores Th17 cells in the oral compartment and triggers a Th2 response.
The results presented are interesting since there is still much that is not known in the treatment of cats infected by FIV, in addition to the fact that it may help to improve the treatments applied to humans. However, there are a number of aspects that remain unclear or need to be improved.
The title does not exactly correspond to the study and the results obtained, since the data correspond to the lymphocyte immunophenotype from blood and lymph nodes but are not related to the pathology of the animals, the lesions, or the evolution of the disease. For example, the increase in CCR6+ CD4+ lymphocytes in the mandibular lymph nodes of FIV cats treated with cART is associated with the rescue of TH17 cells and the restoration of mucosal immunity, which is essential to avoid lentiviral induced oral disease. But data on the clinical conditions of FIV-infected cats, and the presence or absence of oral lesions in c-ART or placebo groups are not provided (physical examination of cats described in lines 103-106), so it is difficult to conclude this only because of the variation in lymphocyte populations.
In the abstract, the authors must include that the cats are experimentally infected with FIV, the age of animals, and that naïve cats are non-infected animals.
Keywords should include immunophenotype and viral loads.
In vivo protocol: It is not indicated that the FIV strain used is type C (line 162). Can it affect the results obtained? It should be discussed in the discussion section.
Figure 1- The text about FIV strain must be located near FIV infection on day 0. Next to the “no therapy” cat, the treatment with the placebo (15% of kleptose) should be added. Naïve are non-infected control cats.
It is not clear why the blood was collected one week prior to FIV inoculation (baseline) and on day 0, but saliva was not collected on day 0 in addition to week 2 and later. Why blood collected at day 0 was not analysed by flow cytometry? (line 185) Why lymph nodes aspirates were not analysed by flow cytometry at day 0? And why these samples were obtained in different weeks than blood samples?
In general, it is difficult to interpret the figures 2-6 in the results section. Significant differences observed (for example, lines 249-250) are not shown in the figures. The effect of cART treatment seems to disappear because at approximately week 30 the values of all groups are similar or identical (Figure 2B 2D, 3F and 3H). In Figure 3E and 3F, some DNA proviral loads in placebo cats are lower than cART-treated cats, and are very similar in RNA viral loads, so it´s no clear that “viral RNA loads in the saliva of cART-treated FIV + cats were lower than placebo treated FIV+ cats” (lines 273 and 277-278) or that “this study demonstrate new evidence of cART inhibiting salivary viral load” (lines 457 and 458). In addition, in other parts of the manuscript, such as the abstract, it is mentioned that there were no significant differences in viraemia in blood and saliva between cats treated with cART or placebo, so it is a bit confusing and the information could be revised and simplified.
Table S1: What is the “treatment” column? The meaning of “F” and “P” must be indicated in the title of the table.
The authors did not observe lymphopenia in cats treated with cART or placebo, could this be associated with the duration of the experiment itself? This could be since they indicate that lymphopenia is seen more frequently in younger animals, chronic infection and clinically ill animals.
The suggestion of a possible relationship between increased CD4+CCR6+ lymphocytes in the mandibular lymph nodes of cART-treated cats and improved oral mucosal immunity is interesting but not supported by animals with or without clinical oral lesions.
Author Response
Response to Reviewers
Reviewer #2
Feline immunodeficiency is an important viral disease in domestic cats caused by chronic and persistent infection by a lentivirus (FIV). At present, there is no definitive therapy to treat infected cats, although several studies have been conducted using different antiretroviral compounds used against human immunodeficiency virus (HIV), none of which have been found to be very effective. Although there is evidence of the action of antiretroviral drugs against FIV both in cell culture and in animal infection, further studies are still lacking to analyse their effect on viral load, clinical and immunopathological parameters of infected animals. On the other hand, very little is known about the use of combination antiretroviral therapies (cART, HAART) in FIV-infected cats similar to what is done in HIV-infected patients.
In the manuscript, the authors evaluate the effect of cART (combined antiretroviral therapy), with three antiretroviral compounds previously used in HIV, in domestic cats experimentally infected with FIV. The main parameters studied are RNA viral and DNA proviral loads in blood and saliva, and lymphocyte immunophenotypes in blood and mandibular lymph nodes. The results are compared with those obtained in infected cats treated with placebo and uninfected cats. The authors conclude that their results seem to suggest that cART inhibits some FIV-induced myelosuppression, restores Th17 cells in the oral compartment and triggers a Th2 response.
The results presented are interesting since there is still much that is not known in the treatment of cats infected by FIV, in addition to the fact that it may help to improve the treatments applied to humans. However, there are a number of aspects that remain unclear or need to be improved:
Thank you so much for the insightful summary and encouraging comments for the manuscript. We have provided a point-by-point response to each comment/concern below.
- The title does not exactly correspond to the study and the results obtained, since the data correspond to the lymphocyte immunophenotype from blood and lymph nodes but are not related to the pathology of the animals, the lesions, or the evolution of the disease. For example, the increase in CCR6+ CD4+ lymphocytes in the mandibular lymph nodes of FIV cats treated with cART is associated with the rescue of TH17 cells and the restoration of mucosal immunity, which is essential to avoid lentiviral induced oral disease. But data on the clinical conditions of FIV-infected cats, and the presence or absence of oral lesions in c-ART or placebo groups are not provided (physical examination of cats described in lines 103-106), so it is difficult to conclude this only because of the variation in lymphocyte populations.
Thank you for your valuable comment. We agree that the narrative is more focused on the immunophenotype than immunopathogenesis, and the title has been amended as suggested. The clinical conditions of FIV-infection have been added as Appendix A (Line 511).
- In the abstract, the authors must include that the cats are experimentally infected with FIV, the age of animals, and that naïve cats are non-infected animals.
Added (Line 20)
- Keywords should include immunophenotype and viral loads.
Thank you for your suggestion. They have been added as recommended (Line 32).
- In vivo protocol: It is not indicated that the FIV strain used is type C (line 162). Can it affect the results obtained? It should be discussed in the discussion section.
Thank you for the opportunity to clarify. Line 91 states that FIVc36 was used for FIV-infection.
- Figure 1- The text about FIV strain must be located near FIV infection on day 0. Next to the “no therapy” cat, the treatment with the placebo (15% of kleptose) should be added. Naïve are non-infected control cats.
Figure 1 has been edited to accommodate this suggestion and include the requested information. The figure caption was also amended to clarify the naïve cat group (Line 112).
- It is not clear why the blood was collected one week prior to FIV inoculation (baseline) and on day 0, but saliva was not collected on day 0 in addition to week 2 and later.
Thank for highlighting an oversight. Baseline blood was collected on day 0, not one week prior to infection. This has been clarified within the text (Line 115). Unfortunately, saliva yields collected at week 0 were not enough for analysis. We were able to improve our sample collection protocols by week 2 to allow for analysis of saliva, but these samples were all we had to work with and we still feel they add value to the study data overall.
- Why blood collected at day 0 was not analysed by flow cytometry? (line 185) Why lymph nodes aspirates were not analysed by flow cytometry at day 0? And why these samples were obtained in different weeks than blood samples?
Flow cytometry was performed at a separate institution. While blood and lymph node samples were collected at baseline (day 0), they were shipped on ice packs obtained from a -80C freezer. Upon receiving the samples for flow cytometry, the cells in all of the samples we completely lysed and no flow cytometry data was able to be obtained. This is the reason that day 0 samples were not analyzed by flow cytometry. We corrected the problem for subsequent collection points. Lymph node aspirates were unable to be obtained in weeks 10 and 15 due to unavailability of the veterinary specialist assigned to the study. These LN aspirates were able to be obtained at weeks 11 and 16 and that data was included.
- In general, it is difficult to interpret the figures 2-6 in the results section. Significant differences observed (for example, lines 249-250) are not shown in the figures. The effect of cART treatment seems to disappear because at approximately week 30 the values of all groups are similar or identical (Figure 2B 2D, 3F and 3H). In Figure 3E and 3F, some DNA proviral loads in placebo cats are lower than cART-treated cats, and are very similar in RNA viral loads, so it´s no clear that “viral RNA loads in the saliva of cART-treated FIV + cats were lower than placebo treated FIV+ cats” (lines 273 and 277-278) or that “this study demonstrate new evidence of cART inhibiting salivary viral load” (lines 457 and 458). In addition, in other parts of the manuscript, such as the abstract, it is mentioned that there were no significant differences in viraemia in blood and saliva between cats treated with cART or placebo, so it is a bit confusing and the information could be revised and simplified.
p-values were added to clarify Figure 2’s significant differences (Line 251, 258).
Figure 3’s caption (Line 276) and the text (Line 280) clarify that the difference in viral RNA loads in saliva were primarily from weeks 0-5, and remained similar onwards.
“this study demonstrate new evidence of cART inhibiting salivary viral load” was an oversight, and it has been removed.
- Table S1: What is the “treatment” column? The meaning of “F” and “P” must be indicated in the title of the table.
“F” is for F-statistic and refers to an F-test used to help calculate the p-value, which is labeled as “P”. Table S1’s caption has been amended to clarity.
- The authors did not observe lymphopenia in cats treated with cART or placebo, could this be associated with the duration of the experiment itself? This could be since they indicate that lymphopenia is seen more frequently in younger animals, chronic infection and clinically ill animals.
We agree with your conclusion, and it is described in Line 360.
- The suggestion of a possible relationship between increased CD4+CCR6+ lymphocytes in the mandibular lymph nodes of cART-treated cats and improved oral mucosal immunity is interesting but not supported by animals with or without clinical oral lesions.
Thank you for your comment. We respectfully believe that the data reasonably supports this statement, which is currently described as speculative.

Reviewer 3 Report
The manuscript entitled “Combination Antiretroviral Therapy and Immunopathogenesis of Feline Immunodeficiency Virus” evaluated that the effects of combination antiretroviral therapy on treating HIV patients in FIV-infected cats. The manuscript is very interesting and has the potential to contribute significantly to the treatment of FIV. However, some points need to be improved.
Major points
1. Please clarify how you decided the doses of drugs which you used.
2. It is written for the Materials and Method that the physical examination of the cat was carried out through an experiment, but please write a result as the result is written nowhere. In addition, please write whether the physical examination. Also, please describe whether the physical examination recording (temperature, weight, hematology, etc.) have improved with the treatment.
3. Because a reader makes it easy to understand it, for example, please describe it as well as figure 1 with figure 1A when you refer to a figure in the text.
4. In Figure 2C, one of the control lines does not seem to start at 0, so check that. Also check placebo and cART in Figure 3E.
5. Maybe it's just that I can't see it, but the placebo in Figure 3E appears to have only 5 lines. Please check it.
6. L286~ CD4+ T cells from cats treated with cART declined cells numbers. Similar results have been seen with CD21+ cells. Please provided evidence that this was not caused by the toxicity of the drugs used.
7. L303~ In the text, it is stated that the number of CD21+ cells collected from the lymph of cART-treated FIV+ cats was lower than that collected from placebo-treated FIV+ cats. On the other hand, according to the legend in Figure S3, the number of CD21+ cells in the cART treated group was low prior to the treatment, but after the treatment there was no difference. The first sentence can be interpreted as a decrease in number due to the treatment, the second sentence can be interpreted as having a difference before the treatment but no longer after the treatment, and the two sentences do not seem to state the same thing. Please correct these two sentences so that they match.
8. In this study, the drugs used in cART did not show direct antiretroviral effects. Please cite and discuss any papers reporting in vitro anti-FIV effects of the drugs you have used. It might suggest that the amount of drug used was not adequate to show direct antiviral activity. Also, if author cite reports that the drugs used act on the cells of the immune system, the results will be more convincing.
Minor points
1. Please unify the abbreviation of the drug in the whole text. For example, Tenofovir disoproxil is TDF in Figure 1 or PMPA in L96.
2. In the legend of Figure 3, it would be easier to understand to write A, B, E, F as proviral DNA and C, D, G, H as viral RNA. The legend also states, "Although salivary viral RNA levels in cART-treated FIV+ cats were lower than placebo-treated FIV+ cats, they remained similar from week 6 onwards." There is, but it is better to write this in the result, not in the legend.
3. L377~ These citations do not appear to indicate an increase in CD4 due to FIV infection. Please check.
4. L379 The reference number 53 is a paper reporting FeLV, not FIV.
Author Response
Response to Reviewers
Reviewer #3
The manuscript entitled “Combination Antiretroviral Therapy and Immunopathogenesis of Feline Immunodeficiency Virus” evaluated that the effects of combination antiretroviral therapy on treating HIV patients in FIV-infected cats. The manuscript is very interesting and has the potential to contribute significantly to the treatment of FIV. However, some points need to be improved.
We appreciate your thorough evaluation and positive suggestions for the manuscript. Please find our response to each improvement points below.
Major points
- Please clarify how you decided the doses of drugs which you used.
Thank you for the opportunity to clarify. The drug doses were based on a cART protocol used to suppress viremia in nonhuman primates infected with simian immunodeficiency virus, and this has been clarified in the Discussion (Line 336).
- It is written for the Materials and Method that the physical examination of the cat was carried out through an experiment, but please write a result as the result is written nowhere. In addition, please write whether the physical examination. Also, please describe whether the physical examination recording (temperature, weight, hematology, etc.)have improved with the treatment.
Thank you for highlighting the oversight. Results related to physical examination were added to Appendix A (Line 511).
- Because a reader makes it easy to understand it, for example, please describe it as well as figure 1 with figure 1A when you refer to a figure in the text.
Thank you for the excellent suggestion. The text has been amended as recommended.
- In Figure 2C, one of the control lines does not seem to start at 0, so check that. Also check placebo and cART in Figure 3E.
Figure 2C and 3E are both accurate. There was not enough blood or saliva collected, respectively, to obtain the data.
- Maybe it's just that I can't see it, but the placebo in Figure 3E appears to have only 5 lines. Please check it.
You are correct. One of the six placebo-treated FIV+ cats was excluded from saliva collection (Line 161).
- L286~ CD4+ T cells from cats treated with cART declined cells numbers. Similarresults have been seen with CD21+ cells. Please provided evidence that this was not caused by the toxicity of the drugs used.
Line 398 describes that changes in CD4+ T cells were similar to previous reports. The changes in CD21+ cells are described in Line 403-407, where the greatest decrease in CD21+ cells occurring prior to cART. Nonetheless, we highlight in Line 408-409 that the cause for these changes in unknown.
- L303~ In the text, it is stated that the number of CD21+ cells collected from the lymph of cART-treated FIV+ cats was lower than that collected from placebo-treated FIV+ cats. On the other hand, according to the legend in Figure S3, the number of CD21+ cells in the cART treated group was low prior to the treatment, but after the treatment there was no difference. The first sentence can be interpreted as a decrease in number due to the treatment, the second sentence can be interpreted as having a difference before the treatment but no longer after the treatment, and the two sentences do not seem to state the same thing. Please correct these two sentences so that they match.
Amended as recommended (Line 308).
- In this study, the drugs used in cART did not show direct antiretroviral effects. Please cite and discuss any papers reporting in vitro anti-FIV effects of the drugs you have used. It might suggest that the amount of drug used was not adequate to show direct antiviral activity. Also, if author cite reports that the drugs used act on the cells of the immune system, the results will be more convincing.
Thank you for this suggestion. We have edited the text in the discussion to clarify and expand on this point staring at line 374.
Minor points
- Please unify the abbreviation of the drug in the whole text. For example, Tenofovir disoproxil is TDF in Figure 1 or PMPA in L96.
Done
- In the legend of Figure 3, it would be easier to understand to write A, B, E, F as proviral DNA and C, D, G, H as viral RNA. The legend also states, "Although salivary viral RNA levels in cART-treated FIV+ cats were lower than placebo-treated FIV+ cats, they remained similar from week 6 onwards." There is, but it is better to write this in the result, not in the legend.
Thank you for the suggestion. A similar statement has been added to the results (Line 280).
- L377~ These citations do not appear to indicate an increase in CD4 due to FIV infection. Please check.
It is unclear which citation(s) specifically the reviewer is referring to since there was no citation in line 377 of the version submitted. However, we suspect the reviewer is referring to citations [6,52–57] in line 380 of the originally submitted version (line 398 of the revision) due to the context of this sentence. We have edited the text and included a citation to support that the FIVC36 strain causes a transient increase in CD4 during acute infection, followed by the gradual decline as typically observed. We believe this should clarify any confusion surrounding this sentence.
- L379 The reference number 53 is a paper reporting FeLV, not FIV.
Thank you for highlighting the oversight. Reference 53 (Quakenbush et al.) was removed.

Round 2
Reviewer 3 Report
The manuscript entitled “Combination Antiretroviral Therapy and Immunophenotype of Feline immunodeficiency Virus” has been revised. Please correct only one of the following minor points.
In Figure 6, please unify the abbreviation of the drug. TEN→EDF、EMT→FTC
Author Response
Figure 6 and the corresponding figure legend were edited as requested to unify abbreviations of the drugs throughout: tenofovir disoproxil fumarate (TDF), emtricitabine (FTC), and dolutegravir (DTG).
Thank you for catching this!